# lncRNA-TCONS_00008552 expression in patients with pulmonary arterial hypertension due to congenital heart disease

Qi Yang[1,2,3☯], Wei Fan[1,2,3☯], Banghui Lai[1,2,3], Bin Liao[1,2,3]*, Mingbin Deng [1,2,3]*

1 Department of Cardiovascular Surgery, Affiliated Hospital of Southwest Medical University, Luzhou, Sichuan Province, China, 2 Metabolic Vascular Diseases Key Laboratory of Sichuan Province, Luzhou, China, 3 Key Laboratory of Medical Electrophysiology, Ministry of Education & Medical Electrophysiological Key Laboratory of Sichuan Province, (Collaborative Innovation Center for Prevention of Cardiovascular Diseases) Institute of Cardiovascular Research, Southwest Medical University, Luzhou, China

☯ These authors contributed equally to this work.
* liaobin@swmu.edu.cn (BL); 594498349@qq.com (MD)

**Data Availability Statement:** All relevant data are within the paper and its Supporting Information files.

## Abstract

Long noncoding RNAs (lncRNAs) are potential regulators of a variety of cardiovascular diseases. Therefore, there is a series of differentially expressed lncRNAs in pulmonary arterial hypertension (PAH) that may be used as markers to diagnose PAH and even predict the prognosis. However, their specific mechanisms remain largely unknown. Therefore, we investigated the biological role of lncRNAs in patients with PAH. First, we screened patients with PAH secondary to ventricular septal defect (VSD) and those with VSD without PAH to assess differences in lncRNA and mRNA expression between the two groups. Our results revealed the significant upregulation of 813 lncRNAs and 527 mRNAs and significant down-regulation of 541 lncRNAs and 268 mRNAs in patients with PAH. Then, we identified 10 hub genes in a constructed protein-protein interaction network. Next, we performed bioinformatics analyses, including Gene Ontology and Kyoto Encyclopedia of Genes and Genomes pathway analysis and subsequently constructed coding-noncoding co-expression networks. We screened lncRNA-TCONS_00008552 and lncRNA-ENST00000433673 as candidate genes and verified the expression levels of the lncRNAs using quantitative reverse-transcription PCR. Although expression levels of lncRNA-TCONS_00008552 in the plasma from the PAH groups were significantly increased compared with the control groups, there was no significant difference in the expression of lncRNA-ENST00000433673 between the two groups. This study bolsters our understanding of the role of lncRNA in PAH occurrence and development and indicates that lncRNA-TCONS_00008552 is a novel potential molecular marker for PAH.

## Introduction

Pulmonary Arterial Hypertension (PAH) is a fatal disease caused by increased blood flow in the pulmonary circulation [1, 2]. The lesion covers the entire vascular layer and is

**Funding:** This work was financially supported by the National Natural Science Foundation of China (82070277), grant recipient (Bin Liao); the Scientific Research Project of Southwest Medical University (2021ZKQN100); grant recipient (Qi Yang); the Scientific Research Project of Southwest Medical University (2021ZKMSO21), grant recipient (Mingbin Deng).

**Competing interests:** The authors have declared that no competing interests exist.

characterised by the gradual increase of pulmonary vascular resistance and pulmonary small vascular plexus lesions, which progressively lead to right-heart hypertrophy and ultimately death due to right-sided heart failure [3]. Currently, there is no specific cure for PAH. The existing molecular targeted drugs, such as endothelin receptor antagonists, phosphodiesterase inhibitors, calcium channel blockers and cyclic prostaglandin analogues, have been reported to achieve certain curative effects in clinical treatment but are unable to curb further deterioration of the condition [4, 5]. PAH pathogenesis is a complex and multifactorial process [6]. Thus, clarification of the molecular mechanism of occurrence and development of PAH and the detection of new molecular markers for early diagnosis, prognosis and therapeutic targets are crucial.

In recent years, long noncoding RNAs (lncRNAs), which are noncoding RNA molecules exceeding 200 nucleotides, have become the focus of medical research [7]. They have been demonstrated as involved in important mechanisms through epigenetic, transcriptional activation, transcriptional inhibition, post-transcriptional regulation, intranuclear transport and other pathways [8–10]. In this complex regulatory role, lncRNAs are mainly involved in the mechanism of competitive endogenous RNA [11]. As a molecular sponge of microRNA (miRNA), lncRNAs indirectly regulate the expression of the miRNA target gene by competitively binding miRNA [3, 12]. The abnormal expression of gene products caused by the change in lncRNA levels is associated with the occurrence and development of cardiovascular disease, which has been confirmed by an increasing number of experimental studies [13–15]. For example, plasma or serum samples from people with atherosclerotic disease show increased levels of lncRNA H19 [16, 17]. Gu et al. [18] identified 185 significantly different lncRNAs in chronic thromboembolic pulmonary hypertension tissues by chip analysis, the most notable of which were NR_036693, NR_027783, NR_033766 and NR_001284. Additionally, lncRNAs induced by platelet-derived growth factor BB modulated the proliferation of pulmonary artery smooth muscle cells (PASMCs) [4]. Furthermore, Liu et al. identified 36 upregulated and 111 downregulated lncRNAs in hypoxic pulmonary artery tissue in a hypoxia-induced PAH model and focused on the role and mechanism of lncRNA TCNS-00034812 in the proliferation and apoptosis of PASMCs [19]. Although several studies suggest that lncRNAs play a crucial role in PAH pathogenesis, the role of lncRNAs in PAH is still in the preliminary stages of research and the specific mechanism remains unelucidated.

Gene chip and high-throughput sequencing technology have rapidly advanced in recent years [20]. Furthermore, the research method of identifying key genes involved in the pathogenesis of disease and developments in bioinformatics have attracted extensive attention from scientific researchers [21, 22].

In this investigation, we used microarray analysis to screen and determine the differential expression profiles of lncRNAs and mRNAs between patients with severe PAH secondary to ventricular septal defect (VSD) and those with VSD without PAH. Then, we analysed the differentially expressed mRNA genes using Gene Ontology (GO) and Kyoto Encyclopedia of Genes and Genomes (KEGG) function enrichment and successfully constructed protein–protein interaction (PPI) and coding-noncoding co-expression (CNC) networks to further explore the potential role of the differentially expressed lncRNAs in PAH. Additionally, quantitative reverse-transcription PCR (qRT-PCR) revealed that the plasma levels of lncRNA-TCONS_00008552 in the PAH groups were significantly increased compared with the controls. Collectively, our findings revealed that lncRNA-TCONS_00008552 played an important role in PAH, thereby enhancing the understanding of the role of lncRNAs in the occurrence and development of PAH.

## Materials and methods

### Ethics statement

We informed the participants about the experimental steps of this study and obtained their consent and signature confirmation. The design of the study was in line with the Helsinki Declaration (http://www.wma.net/en/30publications/10politics/b3/) and was agreed on after discussion by the Ethics Committee of the Affiliated Hospital of Southwest Medical University (No:KY2022209).

### Patients and Samples

Our study cohort included eight children aged 6–10 years who were hospitalised in the Affiliated Hospital of Southwest Medical University, China on June 2022. Four of the children were diagnosed by echocardiography as VSD without PAH (control group, n = 4) and the other four were diagnosed by echocardiography and right cardiac catheterisation as moderate or severe PAH secondary to VSD (PAH group, n = 4). A diagnosis of PAH by right-heart catheterisation was defined as a mean pulmonary arterial pressure >25 mmHg at rest, a pulmonary capillary wedge pressure <15 mmHg and a pulmonary vascular resistance of >3 Wood units. We excluded patients receiving targeted therapy for PAH and those diagnosed with other intracardiac malformations, such as patent ductus arteriosus, large atrial septal defect, or other related conditions, like congenital lung disease, bronchial asthma and congenital pulmonary vascular malformation.

During the cardiac operation, atrial appendage specimens were collected from all patients before cardiopulmonary bypass and blood samples were collected via the jugular vein before performing the midline sternotomy. The plasma and right atrial appendage specimens were then aliquoted and stored at −80°C until RNA extraction.

### RNA isolation and quantitative real-time PCR

We extracted total RNA from each sample using TRIzol reagent (Invitrogen, Carlsbad, USA) according to the manufacturer's instructions. Then, the quantity and quality of extracted RNA were measured with a NanoDrop ND-1000 spectrophotometer (NanoDrop Technologies, USA). RNA integrity was evaluated by standard denaturing agarose gel electrophoresis. The extracted RNAs were reverse-transcribed into cDNAs according to the manufacturer's instructions. The levels of relative gene expression were analyzed by quantitative reverse transcriptase PCR with TB Green™ Premix Ex Taq™ II (TaKaRa, Japan) using the 7500 Real-Time PCR System (Applied Biosystems, USA). The PCR conditions were as follows: initial denaturation at 99°C for 5 min, 30 cycles of 94°C for 20 sec, 60°C for 30 sec, 72°C for 45 sec and final extension at 72°C for 5 min. The blank control did not contain any cDNA. The primer sequences were designed in the laboratory and synthesized by Generay Biotech (Generay, PRC) based on the mRNA sequences obtained from the NCBI database and were listed in S1 Table. The expression levels of lncRNAs were normalized to GAPDH and calculated using the 2−ΔΔCt method [23].

### Expression profiling data and analysis

We used Agilent Feature Extraction software (Agilent, USA) for quantile normalisation of raw data and subsequent data processing. Thereafter, we screened high-quality probes for further analyses. We observed differentially expressed lncRNAs and mRNAs in at least three out of four samples. When comparing the profile differences between two groups (such as disease vs control), we computed the fold change (i.e., the ratio of the group averages) between the

groups for each lncRNA or mRNA. The statistical significance of the difference was estimated using Student $t$-test and lncRNAs or mRNAs with fold change $>2$ and P-value $<0.05$ were considered to show significant differential expression.

## GO and KEGG pathway analysis

The genes were entered into the DAVID database for enrichment analysis of the GO terms and KEGG pathways. The GO enrichment analysis covers three domains: biological process, cellular component and molecular function. A threshold of $P < 0.05$ was considered to indicate significantly enriched GO terms and KEGG pathways; these genes were annotated and the results were visualised.

## PPI network and modular analysis

We entered the top 50 differential genes into the STRING database for protein interaction network analysis to obtain the PPIs. After downloading the PPI network data, we imported them into Cytoscape software, deleted the isolated genes and labelled the upregulated and downregulated genes. The top 10 hub genes in the network were screened using the Cytoscape cyto-Hubba plug-into to enrich and analyse their biological functions.

## CNC network analysis

We calculated Pearson correlation coefficients (PCCs) as co-expression measures to construct gene co-expression networks based on the levels of mRNA and lncRNA expression. The absolute value of parameter PCC > 0.9 and $P < 0.01$ was selected to construct the network and visualise the co-expression networks associated with the top 10 hub genes using Cytoscape v3.7.1 software.

## Statistical analysis

All statistical analyses were performed using SPSS v22.0 (Corp., Armonk, USA). All quantitative data were expressed as the mean ± SD. We used a two-tailed Student $t$-test for comparisons between two groups. A _P-value of $<0.05$ was considered statistically significant.

# Results

## General data

The clinical characteristics of the patient cohort are presented in Table 1. There was no significant difference in age (P = 0.85), sex and weight (P = 0.67) between the two groups (P > 0.05).

**Table 1. Clinical characteristics of the patients.**

| Group | SN | Sex | Weight(Kg) | Age(y) | PASP(mmHg) |
|-------|-----|-----|------------|--------|------------|
| PAH | T2 | F | 22 | 8.2 | 72 |
| | T3 | F | 25 | 7.4 | 62 |
| | T4 | M | 17 | 6.9 | 52 |
| | T6 | F | 18 | 7.5 | 69 |
| CON | C7 | F | 20 | 7.3 | 22 |
| | C9 | M | 20 | 7.6 | 22 |
| | C11 | F | 16 | 6.8 | 21 |
| | C12 | F | 22 | 8.0 | 23 |

Abbreviations: SN: Sample name; PASP: pulmonary artery systolic pressure; PAH: pulmonary arterial hypertension; CON: control groups; F: female; M: male

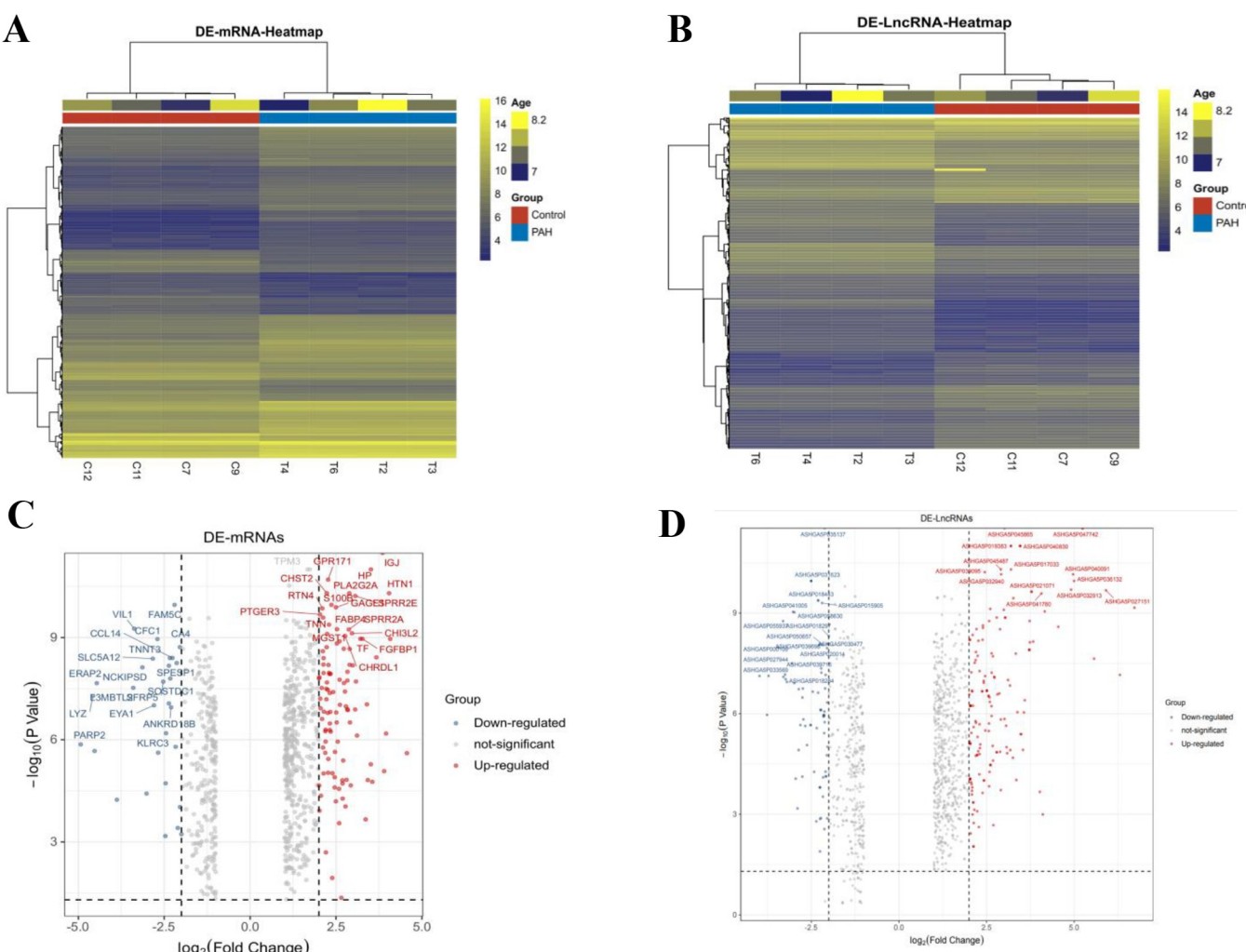

**Fig 1. Differentially expressed profiles of mRNAs and lncRNAs in the PAH groups compared to the control groups using gene microarray analysis.** (A) The hierarchical clustering of differentially expressed mRNAs. n = 4. (B) The hierarchical clustering of differentially expressed lncRNAs; n = 4. (C) Volcano plots representing the variations of mRNAs. (D) Volcano plots representing the variations of lncRNAs.fold change >2, P <0.05.

## Differentially expressed lncRNAs and mRNAs

We screened lncRNAs and mRNAs that were differentially expressed in the PAH and control groups with cut-off criteria of a fold change >2 and adjusted P-value <0.05. We identified 813 upregulated and 541 downregulated lncRNAs and 527 upregulated and 268 downregulated mRNAs in the PAH groups compared with the control groups. Hierarchical clustering and volcano plots clearly showed that the expression patterns of differentially expressed lncRNAs and mRNAs were distinguishable between the two groups (Fig 1).

## Functional enrichment analysis of differentially expressed mRNAs

We used DAVID to analyse GO terms and KEGG pathways to explore the function of the differentially expressed mRNAs. The upregulated mRNAs were enriched in GO terms, including cell-cell signalling, response to toxic substances, positive regulation of interferon-beta production, cytosol, oxygen transporter activity and oxygen binding (Fig 2A), while the downregulated genes were enriched in GO terms, including defence response to bacterium, negative

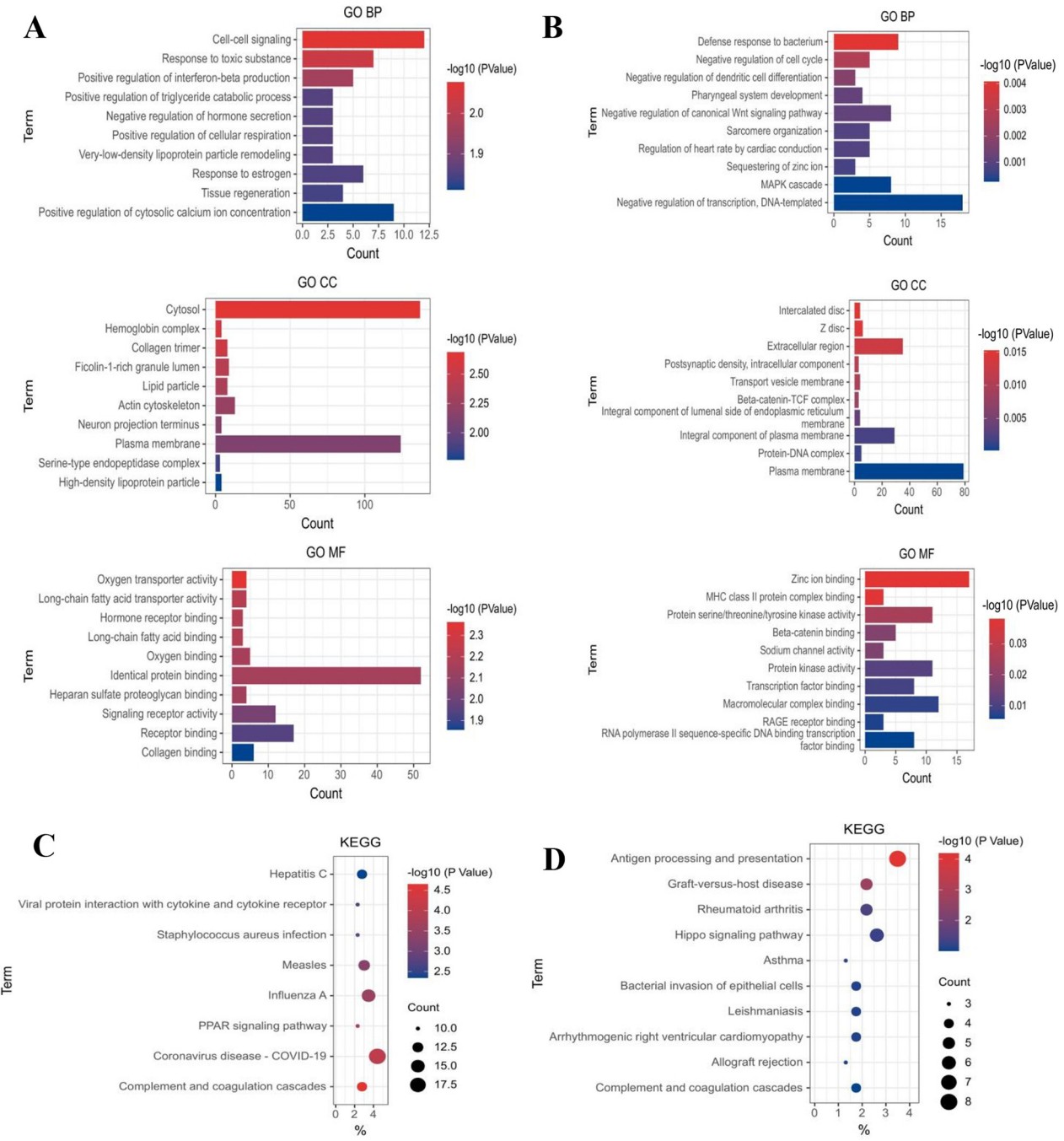

**Fig 2. Functional prediction of differentially expressed mRNAs in PAH.** (A)GO and (C) KEGG pathway analysis of upregulated mRNAs in PAH. (B) GO and (D) KEGG pathway analysis of downregulated mRNAs in PAH. GO, gene ontology; KEGG, Kyoto Encyclopedia of Genes and Genomes.

regulation of cell cycle, extracellular region, zinc ion binding, MHC class II protein complex binding and protein serine/threonine/tyrosine kinase activity (Fig 2B). KEGG pathway analysis revealed that the upregulated mRNAs were mainly involved in complement and coagulation cascades, coronavirus disease 2019, peroxisome proliferator-activated receptor (PPAR) signalling pathway, influenza A and measles (Fig 2C), while the downregulated genes were

involved in antigen processing and presentation, graft-versus-host disease and rheumatoid arthritis (Fig 2D).

## PPI network construction and functional enrichment analysis of hub mRNAs

We selected the top 50 upregulated and downregulated differential mRNAs to construct a PPI network using STRING (Fig 3). We identified 10 hub mRNAs with the highest degree according to the PPI network, which played critical roles in the PPI network (Fig 4A). Then, we

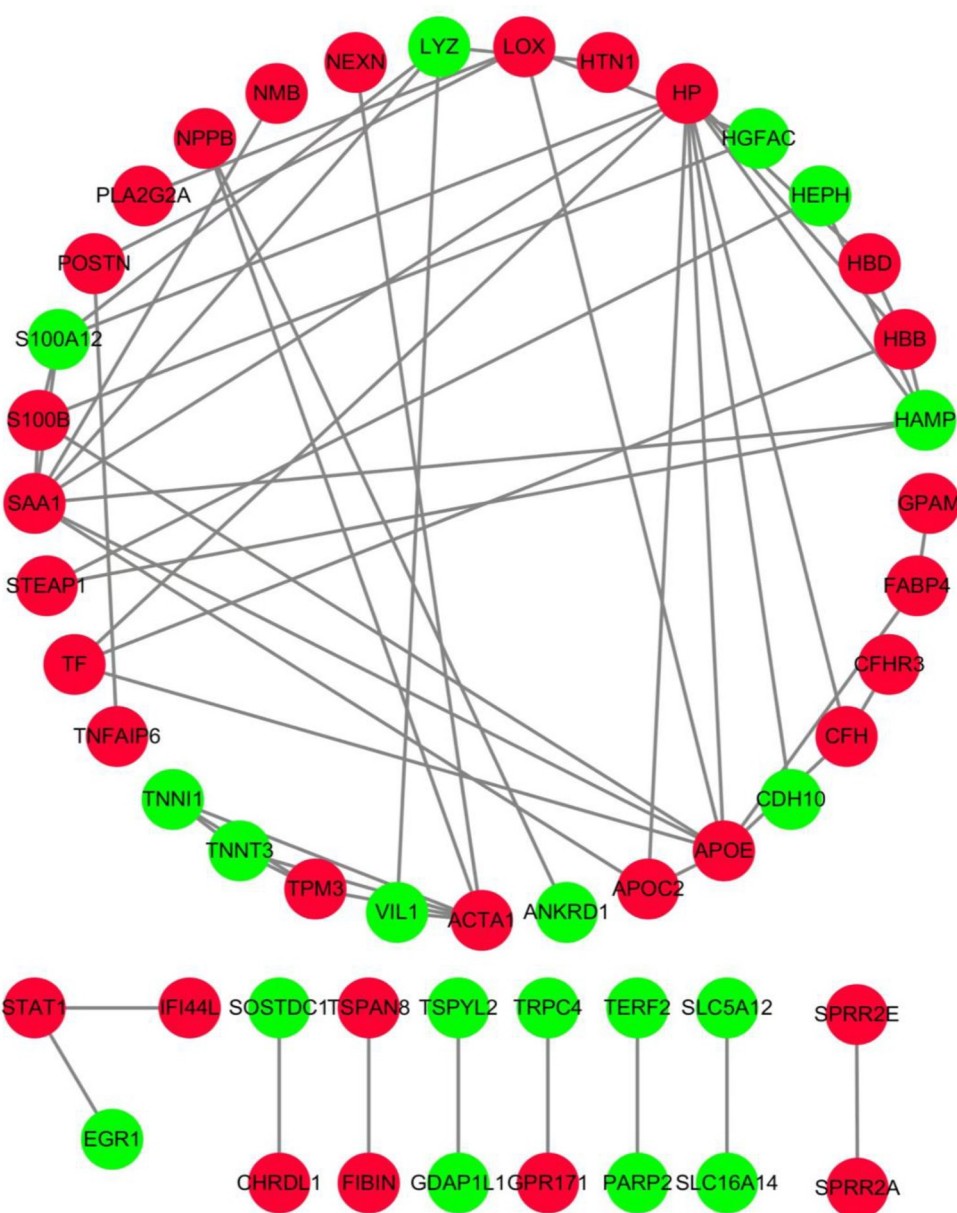

**Fig 3. Construction of protein–protein interaction (PPI) network for upregulated and downregulated differential mRNAs.** Red nodes, upregulated mRNAs; green nodes, downregulated mRNAs.

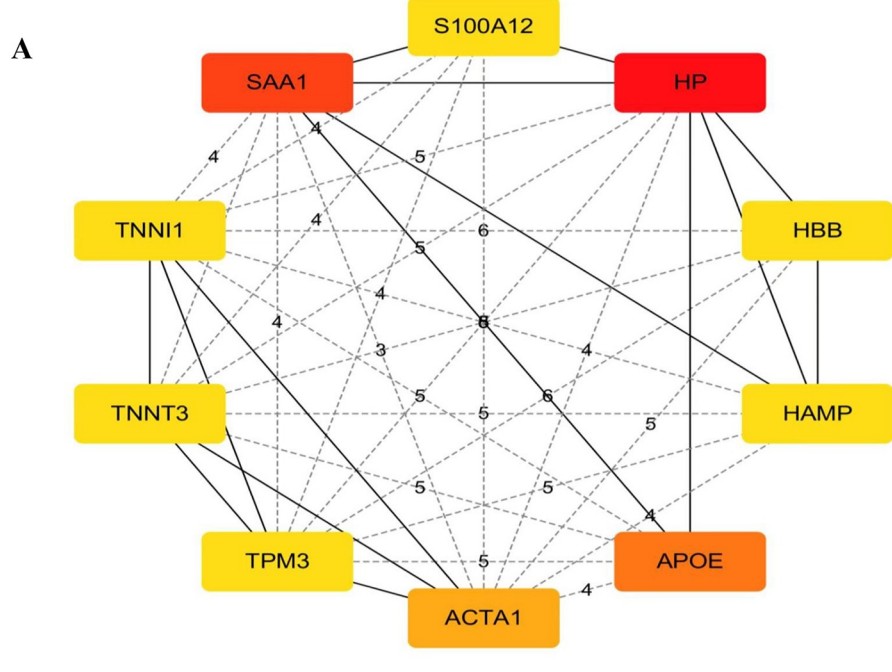

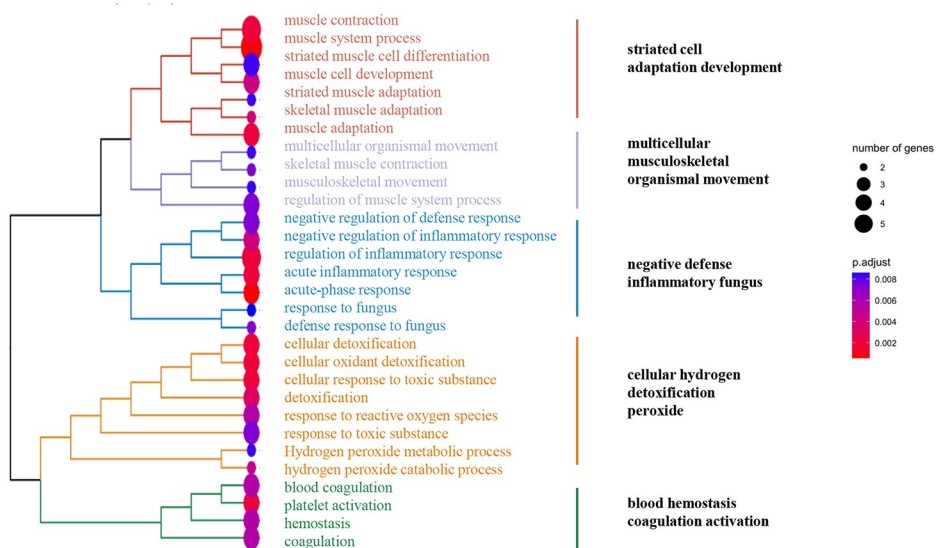

**Fig 4. Identification of hug mRNAs.** (A) It showed that 10 hub mRNAs with the highest degree identified according to the PPI network, which had critical roles in PPI network. (B) Functional prediction of hug mRNAs in PAH.

analysed the GO terms of the hub mRNAs and found that they were predominantly concentrated in striated cell adaptation development, multicellular musculoskeletal organismal movement, negative defense inflammatory fungus, cellular hydrogen detoxification peroxide and blood hemostasis coagulation activation (Fig 4B), which were highly consistent with the pathophysiological process of PAH.

## CNC network construction

To further study the regulatory networks of these hub mRNAs, we constructed an lncRNA-hub mRNA network according to the PCCs (Fig 5). mRNA SAA1 and mRNA TNNT3 had relatively more concentrated functions than the other genes. Next, we analysed the lncRNAs related to the hub mRNAs according to the PCCs. The results revealed that lncRNA-T-CONS_00008552 and lncRNA-ENST00000433673 had the highest correlation coefficient with SAA1 and TNNT3, respectively and were in this network. Therefore, lncRNA-T-CONS_00008552 and lncRNA-ENST00000433673 were selected as candidate genes.

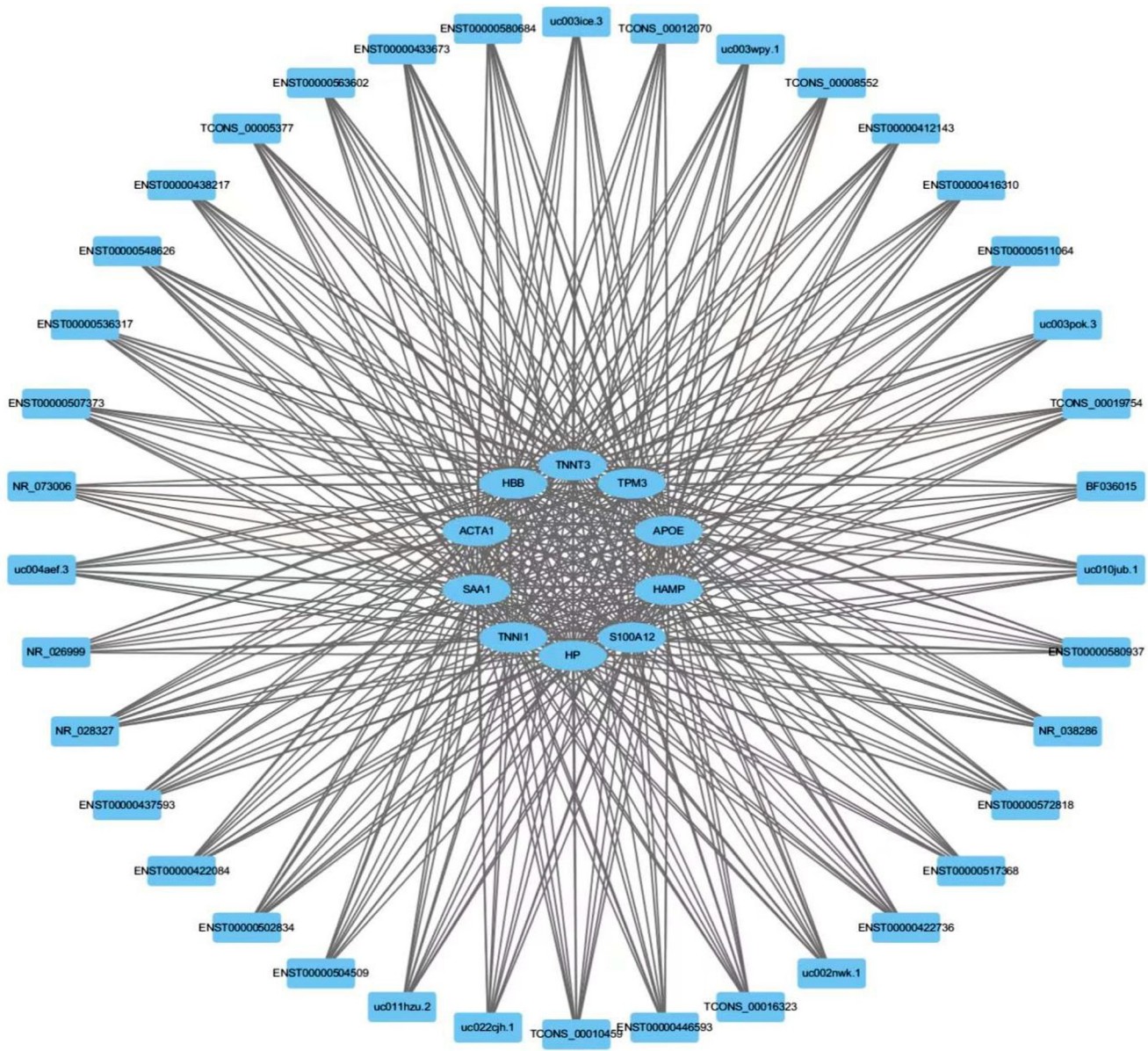

**Fig 5. Construction of lncRNA-mRNA network.** ellipse nodes, hub mRNAs; rectangle nodes, lncRNAs.

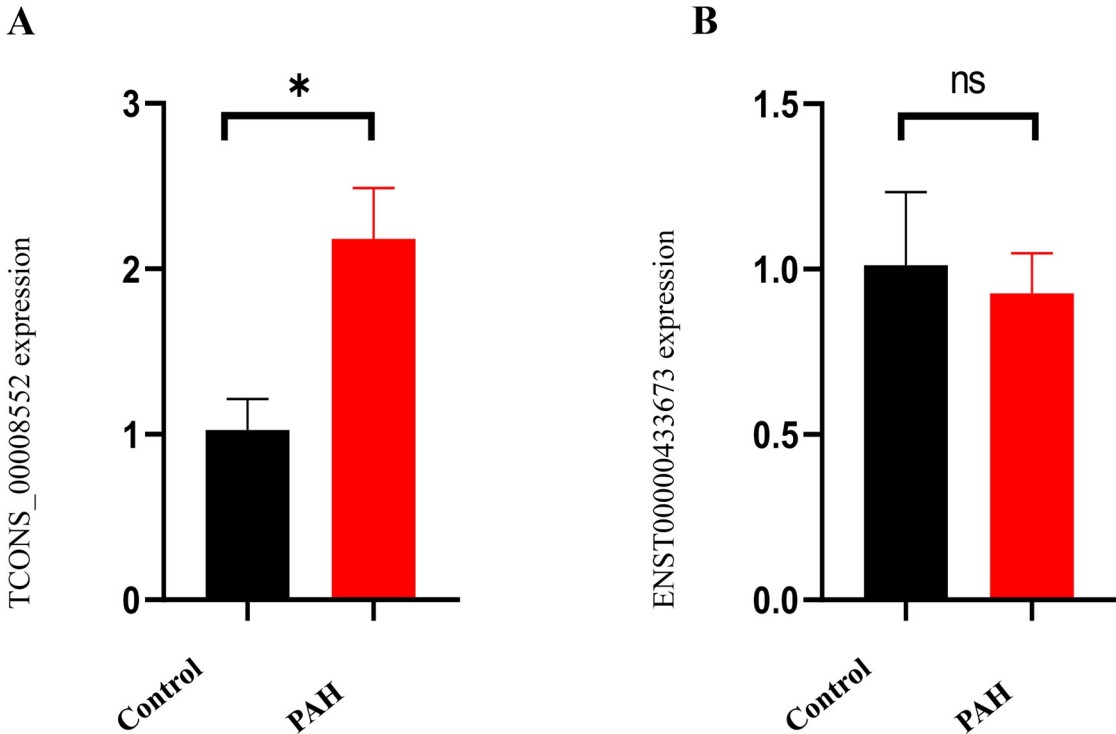

**Fig 6.** qRT-PCR analysis of lncRNA-TCONS_00008552 (A) and lncRNA-ENST00000433673 (B) in the PAH groups compared to the control groups.

### Validation of candidate lncRNAs

We performed qRT-PCR to further validate the candidate lncRNAs using plasma samples from the PAH and control groups (primers for lncRNAs are shown in S1 Table). The levels of lncRNA-TCONS_00008552 in the plasma from the PAH groups were significantly increased compared with the control groups (Fig 6A). Nevertheless, there was no significant difference in the relative expression levels of lncRNA-ENST00000433673 between the two groups (Fig 6B).

## Discussion

PAH is a common and serious complication of congenital heart disease [24, 25]. Although major advances have been made in PAH treatments in recent years, they cannot completely reverse the pathological remodelling and PAH pathogenesis has only been partially elucidated [25]. Therefore, a considerable number of genes and pathways related to PAH need to be studied further. Many studies have reported the contribution of lncRNAs to the underlying pathogenic molecular mechanisms of various human diseases [26, 27], including cardiovascular disease [28, 29]. LncRNAs are mainly involved in the occurrence and development of a variety of diseases by regulating epigenetic transcription and post-transcriptional gene expression [30], but their biological function and mechanism remain unelucidated. The rapid development of molecular medicine in recent years has led to bioinformatics becoming the main method and means to explore disease diagnosis and treatment. Gene chip technology is widely used in various fields because of its high flux, large-scale detection of genes, automation, high sensitivity and rapid turnaround time.

There are a series of differentially expressed lncRNAs in PAH and normal tissues. These differentially expressed lncRNAs may be used as markers to diagnose and even predict the

prognosis of patients with PAH. Sun et al. [31] reported the significant downregulation of *MEG3* in lung and pulmonary arteries of patients with PAH and *MEG3* knockdown affected PASMC proliferation and migration in vitro. Another study demonstrated that lncRNA-UCA1 promoted proliferation and restrained apoptosis by competing with ING5 for hnRNPI in human PASMCs induced by hypoxia, indicating their potential role in curing hypoxic pulmonary hypertension [32].

LncRNAs have become a hot spot in the research of tumour and cardiovascular diseases [33]. However, the lack of systematic research on lncRNA expression in patients with PAH means that the internal relationship between lncRNAs and PAH remains largely unknown [34]. In this study, we first screened the differentially expressed lncRNAs and mRNAs in patients with moderate or severe PAH secondary to VSD and in those with VSD without PAH. We identified 813 upregulated and 541 downregulated lncRNAs and 527 upregulated and 268 downregulated mRNAs in the PAH groups compared with the control groups. These results revealed that PAH pathogenesis was complex and the result of multiple gene interactions. CXCL14 is the significant differential gene, which has an emerging immune and inflammatory modulator mediated cancer response [35]. CXCL14 association with PAH has not been reported; however, CXCL14's key role in inflammation and immune regulation may be a new therapeutic target for pulmonary hypertension [36]. PAPR-2 is another significantly different gene, and studies have shown that the DNA damage/PARP-1 signalling pathway is important for the development of PAH, but the relevance of PAPR-2 to PAH has not been confirmed [37].

GO and KEGG pathway analysis of the differentially expressed mRNAs also provided biological information supporting relationships between genes and gene products in PAH. The results revealed the involvement of differentially expressed genes in a variety of biological processes, cell tissue components and molecular functions, including cell-cell signalling, response to toxic substance, positive regulation of interferon-beta production, cytosol, oxygen transporter activity, oxygen binding, defence response to bacterium and negative regulation of the cell cycle. Further analysis showed that the products of these differentially expressed genes were enriched in a variety of infectious immune pathways and autoimmune pathways, suggesting a close relationship between the immune system and pulmonary hypertension. The substances, reactions and pathways of the above biological processes were mainly consistent with PAH pathophysiology. In terms of KEGG pathway analysis, among the upregulated differentially expressed mRNAs, the biological pathways with high enrichment, such as complement and coagulation cascades, PPAR signalling pathway, influenza A and coronavirus disease 2019, were reported as related to the PAH regulation pathway [24, 38, 39]. In endothelial cell-bone morphogenetic protein type 2 receptor-knockout mice unable to stabilise p53 in endothelial cells under oxidative stress, nutlin-3 rescued endothelial p53 and PPARγ-p53 complex formation and induced target genes, such as *APLN* and *JAG1*, to regenerate pulmonary microvessels and reverse pulmonary hypertension [40]. Zhao et al. demonstrated that miR-27a, PPARγ and ET-1 were cross-inhibited during the pathophysiological processes of PAH, which indicated that the miR-27a/PPARγ/ET-1 signalling pathway was dysregulated. Additionally, bosentan competitively bound to ET-1 receptors and inhibited the miR-27a/PPARγ/ET-1 signalling pathway, thereby delaying PASMC proliferation and affecting PAH development [41]. Our study also suggested that the PPAR signalling pathway was related to PAH regulation and the enrichment of upregulated mRNAs in this pathway was relatively high. In particular, the high enrichment of the measles pathway is naturally worthy of intensive study because it has not been previously reported. Among the enriched biological pathways of downregulated differentially expressed mRNAs, only the rheumatoid arthritis pathway has been proved as involved in PAH. Other pathways had not been reported, including the antigen processing

and graft-versus-host disease pathway with the first enrichment degree and second enrichment degree, warranting further intensive study.

To learn more about the involvement of these genes in PAH, we constructed a differential gene PPI network to further investigate their role, resulting in the identification of 10 core genes. Through the enrichment analysis of GO, the biological processes mainly involved in muscle system process, inflammatory response and cellular detoxification, which was consistent with existing reports. Among them, we found that the functions of TNNT3 and SAA1 were relatively concentrated, although no specific research on their involvement in PAH has been undertaken. Their suitability as potential therapeutic targets should also be investigated in future research.

Finally, we calculated the PCC according to the information on lncRNA and hub mRNA expression. Those pairings with significant correlation were used to construct the lncRNA-hub mRNA regulatory network. These results suggested an association between lncRNAs and mRNAs and revealed that lncRNAs might regulate specific mRNAs and vice versa. Additionally, mRNAs may be directly involved in PAH pathogenesis, while lncRNAs played a role through epigenetic modification of mRNAs. The network analysis revealed that the lncRNA with the highest correlation coefficient was TCONS_00008552 and ENST00000433673. Therefore, we used qRT-PCR to determine the quantity of expressed lncRNA-TCONS_00008552 and lncRNA-ENST00000433673 in plasma samples from the PAH and control groups. The plasma levels of lncRNA-TCONS_00008552 in the PAH groups were significantly increased compared with the control groups. Nevertheless, there was no significant difference in the relative expression levels of lncRNA-ENST00000433673 between the two groups, suggesting lncRNA-TCONS_00008552 is a potential biomarker of PAH and a novel therapeutic target for PAH treatment.

Currently, the molecular mechanisms of lncRNA in PAH mainly focus on proliferation, migration and apoptosis of pulmonary artery smooth muscle cells and dysfunction of pulmonary artery endothelial cells [42]. The mechanism of action of lncRNA-TCONS_00008552 was not explored further because the sequence, structure and function of lncRNAs are poorly conserved across species. Despite this, the lncRNA tests are more complex and expensive. However, lncRNAs can be stably expressed in circulation and have the potential to be used as biomarkers, having a significant impact on the treatment and efficacy determination of PAH.

In conclusion, our study used biochip and bioinformatics technology to reveal differentially expressed lncRNAs and mRNAs in patients with PAH. However, the number of differentially expressed lncRNAs and mRNAs was large and the functions remain unclear. In particular, key lncRNAs and mRNAs require further verification using cell experiments and clinical samples. Our results showed that lncRNA-TCONS_00008552 might act as an activator or inhibitor of genes related to PAH occurrence and development, which warrants further investigation. It is expected that lncRNA-TCONS_00008552 could become a new therapeutic target and biomarker for PAH.

## Supporting information

**S1 Table. Primer sequence for qRT-PCR.**
(XLSX)

**S2 Table. The up-regulated genes in the PAH groups compared to the control groups.**
(XLSX)

**S3 Table. The down-regulated genes in the PAH groups compared to the control groups.**
(XLSX)

**S4 Table. The up-regulated lncRNAs in the PAH groups compared to the control groups.**
(XLSX)

**S5 Table. The down-regulated lncRNAs in the PAH groups compared to the control groups.**
(XLSX)

**S6 Table. The 10 hub genes in a constructed protein-protein interaction network by MCC method.**
(XLSX)

**S7 Table. The lncRNA-mRNA network.**
(XLSX)

## Author Contributions

**Formal analysis:** Wei Fan.

**Funding acquisition:** Bin Liao.

**Investigation:** Qi Yang, Wei Fan.

**Methodology:** Mingbin Deng.

**Resources:** Banghui Lai.

**Software:** Banghui Lai.

**Supervision:** Bin Liao, Mingbin Deng.

**Visualization:** Bin Liao.

**Writing – original draft:** Qi Yang.

**Writing – review & editing:** Mingbin Deng.

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
