## [Decision Letter · Decision Letter 0]

15 Dec 2022

PONE-D-22-20121lncRNA-TCONS_00008552 expression in patients with pulmonary arterial hypertension due to congenital heart diseasePLOS ONE

Dear Dr. Deng,

Thank you for submitting your manuscript to PLOS ONE. After careful consideration, we feel that it has merit but does not fully meet PLOS ONE’s publication criteria as it currently stands. Therefore, we invite you to submit a revised version of the manuscript that addresses the points raised during the review process.

Please discuss the potential diagnostic and therapeutic implications in more detail in the discussion section of your manuscript.

We look forward to receiving your revised manuscript.

Kind regards,

Laszlo Farkas, MD

Academic Editor

PLOS ONE

Journal Requirements:

This work was financially supported by the National Natural Science Foundation of China (82170325), And the Scientific Research Project of Southwest Medical University (2021ZKQN100, 2021ZKMSO21).

However, funding information should not appear in the Acknowledgments section or other areas of your manuscript. We will only publish funding information present in the Funding Statement section of the online submission form. 

This work was financially supported by the National Natural Science Foundation of China (82170325), grant recipient (Bin Liao); the Scientific Research Project of Southwest Medical University (2021ZKQN100); grant recipient (Qi Yang); the Scientific Research Project of Southwest Medical University (2021ZKMSO21), grant recipient (Mingbin Deng)

This work was financially supported by the National Natural Science Foundation of China (82170325), grant recipient (Bin Liao); the Scientific Research Project of Southwest Medical University (2021ZKQN100); grant recipient (Qi Yang); the Scientific Research Project of Southwest Medical University (2021ZKMSO21), grant recipient (Mingbin Deng)

This article had not been published previously, and is not under consideration for publication elsewhere. The authors declared that no competing interests existed,and the publication was approved by all authors. The authors declared that it will not be published elsewhere in the same form, in English or in any other language, including electronically without the written consent of the copyright-holder if this article was accepted.

7. Please include your tables as part of your main manuscript and remove the individual files. Please note that supplementary tables (should remain/ be uploaded) as separate "supporting information" files

Reviewers' comments:

Reviewer's Responses to Questions

**Comments to the Author**

1. Is the manuscript technically sound, and do the data support the conclusions?

Reviewer #1: Yes

Reviewer #2: Yes

2. Has the statistical analysis been performed appropriately and rigorously? 

Reviewer #1: Yes

Reviewer #2: Yes

3. Have the authors made all data underlying the findings in their manuscript fully available?

Reviewer #1: Yes

Reviewer #2: Yes

4. Is the manuscript presented in an intelligible fashion and written in standard English?

Reviewer #1: Yes

Reviewer #2: Yes

5. Review Comments to the Author

Reviewer #1: Overall, this is an excellent and thorough manuscript. Considering that previous research has already shown a positive correlation of the expression of lncRNAs in CHD embryology in general. For example, in TOF cases without documented PAH. This manuscript is novel in identifying genes lncRNA-TCONS_00008552 and lncRNA-ENST00000433673, specific to PAH in CHD.

Reviewer #2: The methodology is correct and article is well written. No additional statistics is required. The article is intersting and clinical relevance is discussed. The statistics performed is also good. Perhaps, the future potential of the article can be written.

6. PLOS authors have the option to publish the peer review history of their article (what does this mean?). If published, this will include your full peer review and any attached files.

Reviewer #1: **Yes: **Dr. Samaa Akhtar

Reviewer #2: **Yes: **Mark Christopher Arokiaraj

---

## [Author Response · Author response to Decision Letter 0]

23 Dec 2022

Thank you for your letter and the reviewers' comments concerning our manuscript entitled " lncRNA-TCONS_00008552 expression in patients with pulmonary arterial hypertension due to congenital heart disease" (PONE-D-22-20121). Those comments are pretty valuable and helpful for revising and improving our paper, as well as the important guiding significance to our research. We have studied the comments carefully and have made corrections which we hope meet with approval. Revised portions are marked in red in the Revised Manuscript with Track Changes.

---

## [Editor Report · Decision Letter 1]

17 Jan 2023

lncRNA-TCONS_00008552 expression in patients with pulmonary arterial hypertension due to congenital heart disease

PONE-D-22-20121R1

Dear Dr. Deng,

We’re pleased to inform you that your manuscript has been judged scientifically suitable for publication and will be formally accepted for publication once it meets all outstanding technical requirements.

Kind regards,

Laszlo Farkas, MD

Academic Editor

PLOS ONE
---

## [Editor Report · Acceptance letter]

19 Jan 2023

PONE-D-22-20121R1 

lncRNA-TCONS_00008552 expression in patients with pulmonary arterial hypertension due to congenital heart disease 

Dear Dr. Deng:

I'm pleased to inform you that your manuscript has been deemed suitable for publication in PLOS ONE. Congratulations! Your manuscript is now with our production department. 

Kind regards, 

on behalf of

Dr. Laszlo Farkas 

Academic Editor

PLOS ONE